# A flexible ultrasensitive optoelectronic sensor array for neuromorphic vision systems

Qian-Bing Zhu[1,2,9], Bo Li[1,2,9], Dan-Dan Yang[3], Chi Liu [1], Shun Feng[1,4], Mao-Lin Chen [1], Yun Sun[1], Ya-Nan Tian[5], Xin Su[6], Xiao-Mu Wang [6], Song Qiu [7✉], Qing-Wen Li[7], Xiao-Ming Li[3✉], Hai-Bo Zeng [3], Hui-Ming Cheng [1,2,8✉] & Dong-Ming Sun [1,2✉]

The challenges of developing neuromorphic vision systems inspired by the human eye come not only from how to recreate the flexibility, sophistication, and adaptability of animal systems, but also how to do so with computational efficiency and elegance. Similar to biological systems, these neuromorphic circuits integrate functions of image sensing, memory and processing into the device, and process continuous analog brightness signal in real-time. High-integration, flexibility and ultra-sensitivity are essential for practical artificial vision systems that attempt to emulate biological processing. Here, we present a flexible optoelectronic sensor array of 1024 pixels using a combination of carbon nanotubes and perovskite quantum dots as active materials for an efficient neuromorphic vision system. The device has an extraordinary sensitivity to light with a responsivity of $5.1 \times 10^7$ A/W and a specific detectivity of $2 \times 10^{16}$ Jones, and demonstrates neuromorphic reinforcement learning by training the sensor array with a weak light pulse of $1\,\mu W/cm^2$.

[1] Shenyang National Laboratory for Materials Science, Institute of Metal Research, Chinese Academy of Sciences, Shenyang, China. [2] School of Material Science and Engineering, University of Science and Technology of China, Hefei, China. [3] College of Materials Science and Engineering, Nanjing University of Science and Technology, Nanjing, China. [4] School of Physical Science and Technology, ShanghaiTech University, Shanghai, China. [5] College of Information Science and Engineering, Northeastern University, Shenyang, China. [6] School of Electronic Science and Engineering, Nanjing University, Nanjing, China. [7] Suzhou Institute of Nano-Tech and Nano-Bionics, Chinese Academy of Sciences, Suzhou, China. [8] Tsinghua-Berkeley Shenzhen Institute, Tsinghua University, Shenzhen, China. [9]These authors contributed equally: Qian-Bing Zhu, Bo Li. ✉email: sqiu2010@sinano.ac.cn; lixiaoming@njust.edu.cn; cheng@imr.ac.cn; dmsun@imr.ac.cn

The human visual system is essential for both survival and learning. It is an efficient process in which the retina detects light stimuli and pre-processes image information in parallel before the brain conducts more complex actions[1–3]. In recent years, digital vision systems, based on conventional complementary metal-oxide-semiconductor (CMOS) imagers or charge-coupled device (CCD) cameras[4–9], have been rapidly developed to achieve the computer vision through extended interfaced digital processing units on serial or coarsely parallel structures[10–13]. However, these conventional digital artificial vision systems tend to consume a lot of power, and have a large size and high cost for practical applications, and neuromorphic vision sensors inspired by biological systems that integrate image sensing, memory, and processing are expected to overcome these disadvantages[14–18].

For the development of a high-performance neuromorphic vision system, optoelectronic sensors with ultra-high responsivity, detectivity, and signal-to-noise ratio are necessary to offer enhanced imaging capability under extreme dim light conditions[19]. For the selection of an active sensing material, the all-inorganic perovskite $CsPbBr_3$-QDs have excellent optoelectronic response performance[20–22], and CNTs can significantly improve the detection signal-to-noise ratio of the sensor due to the excellent carrier mobility and on/off ratio[23–26]. Both materials can be fabricated into uniform large-area films with excellent flexibility and stability, and the combination of these two materials provides a new strategy for the design and fabrication of high-performance neuromorphic vision sensors.

Here, we report a flexible optoelectronic sensor array with 1024 pixels using a combination of CNTs and $CsPbBr_3$-QDs as the active materials, which not only shows an extraordinary sensitivity to light but also has information storage and data preprocessing ability. The device shows a high responsivity of $5.1 \times 10^7$ A/W and an ultra-high specific detectivity of $2 \times 10^{16}$ Jones. It is also the first time that neuromorphic reinforcement learning has been experimentally demonstrated by training a highly integrated sensor array with a weak light pulse of 1 µW/cm². Similar to biological systems, the photoreceptor, memory element, and computational node components share the same physical space in the array and process the information in parallel and in real-time, which makes them attractive for constructing artificial vision systems that attempt to emulate biological processing.

## Results

### Device design and characterization.
Figure 1a shows a schematic of the design of the phototransistor with a buried-gate structure, where the channel consists of high-purity (>99.9%) semiconducting CNTs (Supplementary Figs. 1–3) and perovskite $CsPbBr_3$-QDs (Supplementary Fig. 4) that respectively act as active materials for electrical transport and photon absorption ('Methods' and Supplementary Figs. 5 and 6). The CNT random network covered by the uniformly-dispersed $CsPbBr_3$-QDs (Fig. 1b, c) to ensure the formation of the high-quality CNT/$CsPbBr_3$-QD interface and gives a uniform device performance. Figure 1d shows typical transfer characteristics ($I_{DS}$–$V_{GS}$) of the phototransistor in the dark and under various lighting power densities ($P$). The strong photo-response observed shows a largely positive shift in the transfer characteristic curves as the lighting power density increases, and a maximum off-current ratio of $3.6 \times 10^6$ has been achieved for the dark and illuminated conditions (Supplementary Fig. 7). But channel materials composed only of $CsPbBr_3$-QD or CNT cannot meet the high optical response requirements (Supplementary Fig. 8). The output ($I_{DS}$–$V_{DS}$) characteristics at various lighting power densities for fixed $V_{GS} = 5$ and 0 V are also measured (Supplementary Fig. 9),

indicating that the optoelectronic performance of the phototransistor is related to the lighting power density.

Figure 1e shows the energy band diagram at the CNT/$CsPbBr_3$-QD interface for the photogating mechanism. The top panel shows that a built-in electric field that equilibrates the Fermi levels was formed, which leads to band bending at the interface due to the energy band mismatch (Supplementary Fig. 10)[27,28], contributing to the negative shift observed in the dark of the transfer characteristics after spin-coating $CsPbBr_3$-QDs on the CNTs (Supplementary Fig. 11). For the light-on state in the bottom panel, a highly effective dissociation of photogenerated electron-hole pairs occur at the interface between the CNTs and the QDs, as shown in the steady-state and transient photoluminescence (PL) spectra (Supplementary Fig. 12). The longer the exciton lifetime of the $CsPbBr_3$-QDs, the stronger its fluorescence intensity. Therefore, the exciton separation at the CNT/$CsPbBr_3$-QD interface is faster than that of the $CsPbBr_3$-QDs, which is related to the shorter exciton lifetime. The holes separated by the built-in electric field are transferred from the valence band of the QDs to that of the CNTs, and the electrons remain trapped in the $CsPbBr_3$-QDs. Therefore, the negatively-charged QDs induce positive carriers in the CNT film through capacitive coupling that shifts transfer curves in the positive direction[29,30].

### Optoelectronic characteristics.
Figure 2 shows the detailed optoelectronic performance to show the figures of merit of the phototransistor. The responsivity ($R$) decreases with increasing $P$ because of the saturated absorption when $P$ is large, reaching the maximum value of $5.1 \times 10^7$ A/W at the minimum $P$ of 0.01 µW/cm² (Fig. 2a). The dependence of the responsivity on $P$ under various $V_{GS}$ of −5, 0, and +5 V are measured, as shown in Supplementary Fig. 15. It can be found that the photocurrent and responsivity are very similar, especially in the cases of higher lighting power densities, which indicates that the negatively-charged QDs are the dominant factor leading to the increase in current. The external quantum efficiency (EQE) also shows a similar downward tendency and reaches the highest value of $1.6 \times 10^{10}$%. As a figure of merit used to characterize performance, the specific detectivity ($D^*$) of $2 \times 10^{16}$ Jones is achieved in Fig. 2b, owing to the ultrahigh on- and off-current ratio and the high response to weak light with a wavelength of 405 nm (the case of 516 nm, in Supplementary Fig. 17). To benchmark our device, we compared its performance with those of devices constructed of various low-dimensional (0D[28,31,32], 1D[33,34], and 2D[35–37]) materials, organic[38,39], and hybrid[27,40–48] materials in Fig. 2c, and our device shows an ultra-high detectivity for reported devices made with various materials and structures and a comparable responsivity to the highest value for graphene-PbS QDs[40].

Figure 2d shows the switching characteristics of the phototransistor illuminated by a 516 nm light pulse and triggered by applying a short gate pulse voltage from +5 to 0 V to cause a discharge of trapped charge carriers[40,49,50], which results in a rapid decay of photocurrent. The measured response indicates a rise time of 3.3 ms and a fast decay time of 1.1 ms at 0.78 W/cm² (Supplementary Figs. 18–20). In addition, the fabricated device shows long-term stability after being stored in ambient air for more than 8 months (Supplementary Fig. 21) and excellent flexibility when stress is applied, and the $I_{DS}$–$V_{GS}$ curves are almost identical for bending strains ($\varepsilon$) from 0 to 0.4% (Supplementary Figs. 22 and 23).

### Light-tunable synaptic characteristics.
The phototransistor shows memory characteristics and light-dose-dependent response characteristics (Supplementary Figs. 24 and 13), which allow us to mimic the basic features of synaptic plasticity in emulating the

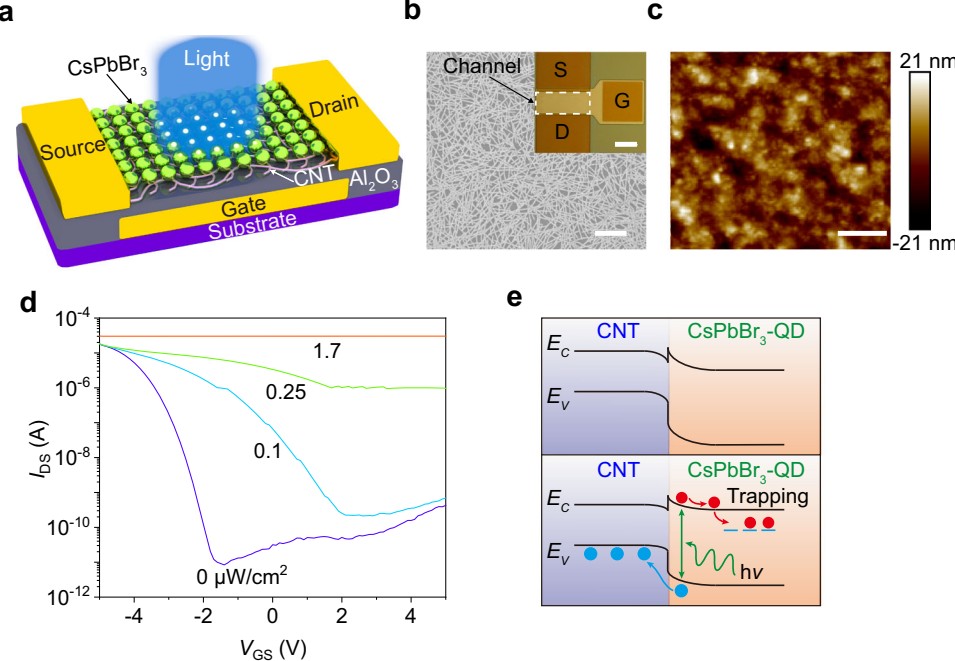

**Fig. 1 Device design and characterization. a** Schematic of the phototransistor with a CNT/CsPbBr$_3$-QD channel. **b** Scanning electron microscope (SEM) image of a CNT film (scale bar, 1 μm). Inset: optical microscope image of the fabricated device (scale bar, 50 μm). **c** Atomic force microscope (AFM) image of a CsPbBr$_3$-QD film (scale bar, 250 nm). **d** Room temperature transfer characteristics ($I_{DS}$ – $V_{GS}$) of the device at $V_{DS} = 1$ V using a collimated incident beam of laser light with a wavelength $λ$ of 516 nm and power densities ($P$) increasing from 0 to 1.7 μW/cm$^2$. **e** Energy band diagram at the light-off (top panel) and light-on states (bottom panel).

learning and memory functions of the human brain, including a transition from short-term plasticity (STP) to long-term plasticity (LTP) (Supplementary Figs. 25 and 26). The effect of paired-pulse facilitation (PPF) among the basic synaptic characteristics is demonstrated in this device by applying two successive optical pulses with a $P$ of 48 μW/cm$^2$, a pulse width of 20 ms, and different pulse intervals, as shown in Fig. 2e. The PPF ratio is defined by the ratio of A$_2$ to A$_1$, where A$_1$ and A$_2$ are, respectively, the peak amplitudes of the current from the first and second optical pulses. When the optical pulse is applied to the phototransistor, the current in the CNT film channel increases and the trapped photogenerated electrons inside the perovskite QD layer take a long time to decay. After application of the second photonic pulse, a higher internal electric field as a result of more trapped electrons induces a higher current level. Furthermore, the PPF index decreases gradually when the pulse interval increases, and a 1 s interval generates a PPF index of more than 180%. Figure 2f shows the induction of long-term potentiation during 500 light pulses at various lighting power densities to demonstrate the persistent strengthening of synaptic characteristics, which means that the optical signal was gradually learned and remembered by our phototransistor. The current increases steadily with an increase in the number of pulse stimulations, and the number of optical pulses needed to achieve a target synaptic weight is reduced for a higher lighting power density and speeds up the associated learning process.

**Flexible optoelectronic sensor array.** Figure 3 shows a 32 × 32 sensor array and its functional demonstration in a neuromorphic vision system. From macro- to micro-levels, the photographic images (Fig. 3a–c and Supplementary Fig. 28) show the array chip mounted on a printed circuit board (PCB), the bonding wires and interconnections in the circuit, and the individual sensor unit with a similar construction to the afore-introduced phototransistor. All devices with 1024 pixels in the

sensor array have been tested and the device yield is 100%, demonstrating the excellent performance uniformity of the device (Supplementary Fig. 29), which is crucial for high-quality image sensing ability.

It was found that the on-current of a single phototransistor increased steadily with an increase in the pulse number of light (Fig. 2f), and the pulse number was reduced for a higher lighting power density to achieve the target synaptic weight, thereby representing the persistent strengthening of synaptic characteristics. The on-currents of 1024 pixels in the sensor array have excellent uniformity, which enables high-quality image sensing, and the larger on-current represents a deeper impression in the evolution of learning and training of images, which allows us to demonstrate the function of neuromorphic pattern reinforcement. This behavior is similar to human vision, where the features of familiar faces are clearer than the features of a strange face occasionally seen (3d). After training 0, 10, 20, 50, 100, and 200 pulses with an ultra-weak light (1 μW/cm$^2$), the weight map of the sensor array obtained shows different resemblance degrees, as well as image sharpness to the input number 8 pattern (Fig. 3e). The calculated accuracy between the ideal input picture and the trained weight map increases with the increase of the number of training pulses, and reached 95% for the case of 200 pulses (Supplementary Fig. 34). In addition, Fig. 3f shows the weight map of the sensor array after training with 10 pulses under a 405 nm light with various lighting power densities, indicating that the higher lighting density can speed up the pattern learning process. Therefore, by training a highly integrated sensor array with weak light pulses, the function of neuromorphic reinforcement learning has been demonstrated experimentally. This is similar to what happens in interpersonal communication, that is, the more you deal with someone, the more facial features you know. We carried out a simulation to mimic the evolution of the learning process of a human face (Fig. 3g) on the basis of the experimental synaptic characteristics in Fig. 3e (Supplementary

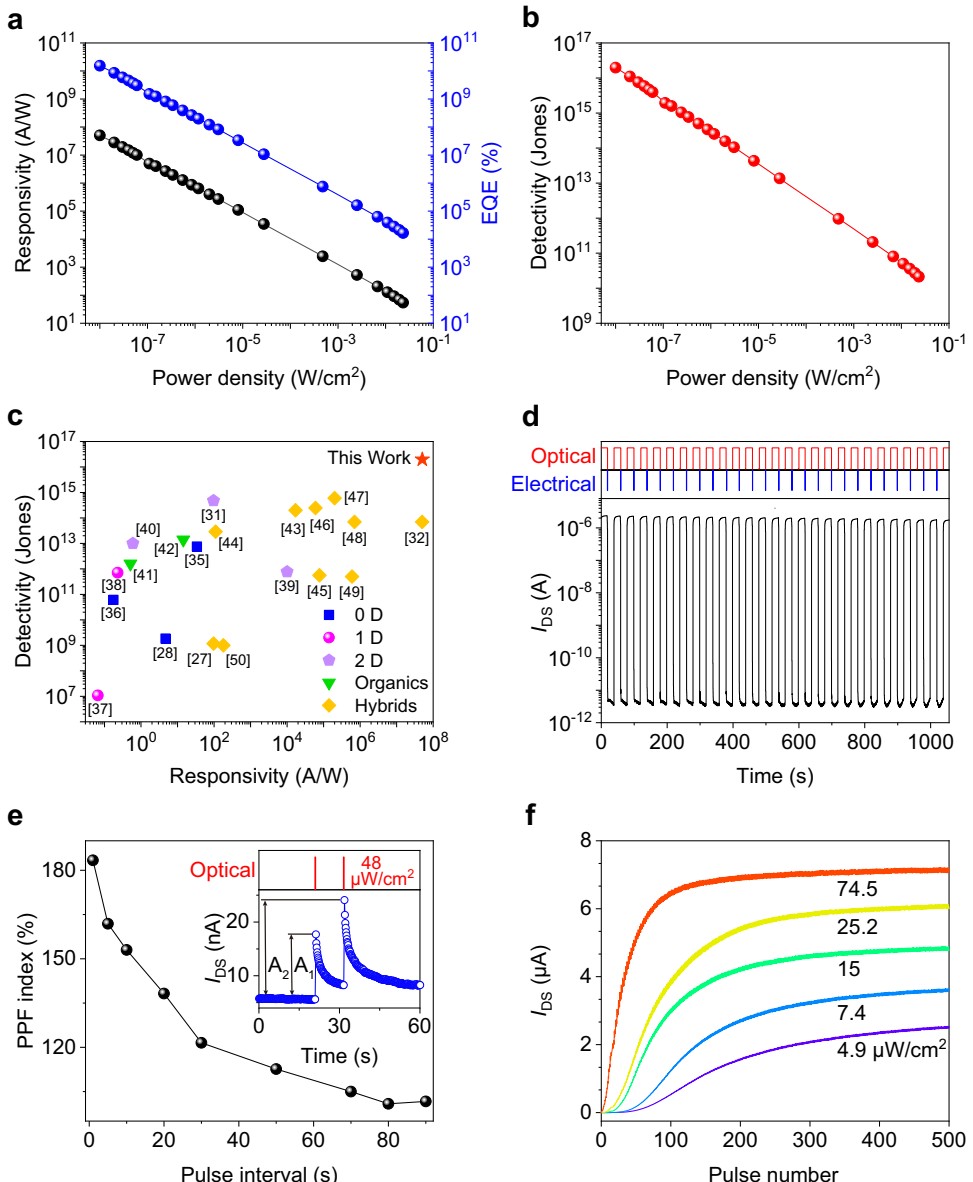

**Fig. 2 Optoelectronic and synaptic characteristics. a** Dependence of the responsivity ($R$) and the external quantum efficiency (EQE) on the lighting power density ($P$). $R = I_{ph}/P(L_{ch} \times W_{ch})$, where $I_{ph}$ is the photocurrent, $L_{ch}$ and $W_{ch}$ are, respectively, the channel length (20 μm) and channel width (100 μm). EQE $= hcR/e\lambda$, where $h$ is the Planck constant, $c$ the speed of light, and $e$ the electron charge. $\lambda = 405$ nm. **b** Dependence of the specific detectivity ($D^*$) on the $P$. $D^* = R(L_{ch} \times W_{ch})^{1/2}/(S_n)^{1/2}$, where $S_n$ is the noise power density (Supplementary Fig. 16). **c** Benchmark of the device performance demonstrating an ultra-high detectivity among reported devices made using various materials and structures. **d** Switching characteristics of the device under a 516 nm light with a $P$ of 0.78 W/cm$^2$ and a reset voltage pulse (+5 to 0 V, pulse width 100 ms) to the gate electrode. $V_{DS} = 1$ V, $V_{GS} = 5$ V. **e** PPF index decreases gradually when the pulse interval increases. Inset: PPF achieved by two successively applied optical pulses (48 μW/cm$^2$, pulse width 20 ms, pulse interval 10 s). **f** Long-term potentiation with 500 optical pulses (pulse width, 20 ms; pulse interval, 500 ms) at various lighting power densities.

Figs. 35 and 36), and verified that more facial features are learned as the number of training pulses increases (Supplementary Fig. 37).

## Discussion

We have demonstrated a flexible optoelectronic sensor array with special features including a high integrated density of 1024 pixels, ultra-sensitivity with an ultra-high specific detectivity for visible light, and the ability for both image sensing and bio-inspired information processing. The active channel consisting of semiconducting carbon nanotubes and perovskite CsPbBr$_3$ quantum dots plays the key roles in the photogenerated carrier separation and transport, achieving a high responsivity of $5.1 \times 10^7$ A/W and

an ultra-high specific detectivity of $2 \times 10^{16}$ Jones. The optoelectronic sensor simultaneously acts as an artificial photoreceptor and a biological synapse, and, thus, directly responds to optical stimuli and performs light-tunable synaptic plasticity for functional pre-processing. These results provide motivation for the development of artificial neuromorphic visual systems to simulate the flexibility, complexity, and adaptability of biological vision systems.

## Methods

**Preparation of semiconducting CNTs and CsPbBr$_3$-QDs.** The raw arc-discharge CNT purchased from Carbon Solution Inc. (https://carbonsolution.com/products/ap-swnt) has a narrow diameter distribution with a peak value of 1.55 ± 0.1 nm and a bundle length of 1–5 μm. High-purity (>99.9%) semiconducting CNTs were sorted by mixing bulk CNTs with a dispersant 9-(1-octylonoyl)-9H-carbazole-2,7-

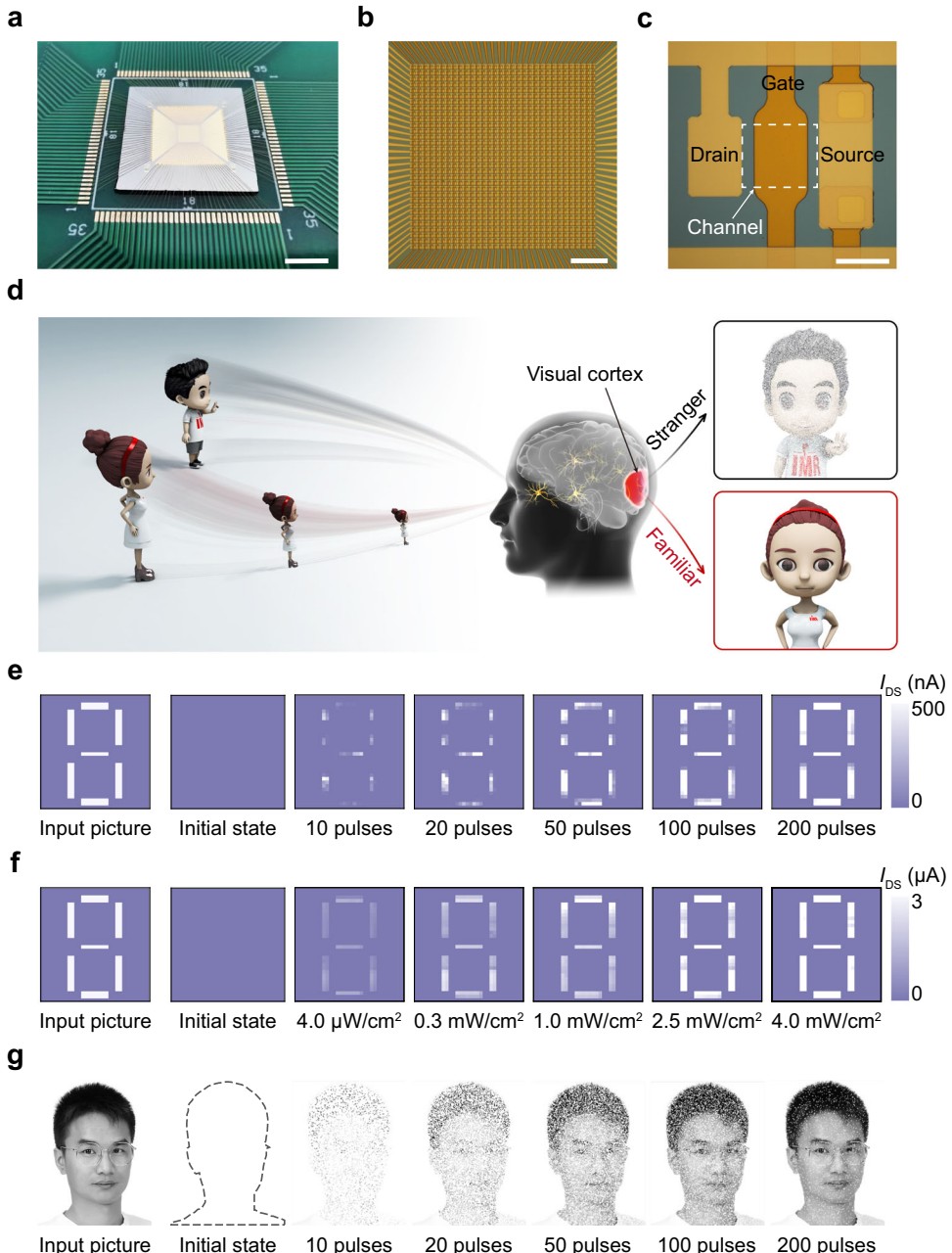

**Fig. 3 Optoelectronic sensor array. a** A sensor array chip with wire bonding on a PCB (scale bar, 5 mm). **b** Optical micrograph of a 32 × 32 sensor array (scale bar, 500 μm). **c** Magnified image of an individual sensor unit with a channel dimension of 20 × 20 μm² (scale bar, 20 μm). **d** Schematics of the impression of human visual systems when strange and familiar faces are observed. **e** Measured training weight results of a number 8 pattern in the initial state and after training with 10, 20, 50, 100, and 200 pulses under 405 nm light with a lighting power density of 1 μW/cm² (pulse width, 250 ms; pulse interval, 250 ms). **f** Measured training weight results of the sensor array after training with 10 pulses under a 405 nm light with various lighting power densities of 4.0 μW/cm², 0.3 mW/cm², 1.0 mW/cm², 2.5 mW/cm², and 4.0 mW/cm² (pulse width, 250 ms; pulse interval, 250 ms). **g** Simulation results of a man's face in the initial state and after training processes.

diyl (PCz) in a xylene solution, followed by ultrasonic stirring for 30 min, and then centrifugation at 45,000 g for 1 h to remove CNT bundles and insoluble substances. The supernatants were collected for use as the channel materials[51]. For the preparation of the CsPbBr₃-QDs, PbBr₂, 1-octadecene, oleic acid, and oleylamine were loaded into a three-neck flask and degassed for 30 min at 120 °C under an Ar flow. After complete dissolution of the PbBr₂, the temperature was increased to 170 °C where it was maintained for 30 min under an Ar atmosphere. The preheated Cs-oleate solution was swiftly injected into the transparent precursor solution for a 5 s reaction, and the mixture was cooled using a water bath. Ethyl acetate was then added to the crude solution with a volume ratio of 1:3 and the mixture was centrifuged at 16,000 g for 1 min, and the procedure was repeated once for better purification, and the final CsPbBr₃-QDs were dispersed in hexane to obtain a clear solution[52].

**Device fabrication on a rigid substrate**. Gate electrodes (Ti/Au: 5/50 nm) were fabricated by standard photolithography, electron-beam evaporation (EBV), and lift-off processes. A 40-nm-thick Al₂O₃ dielectric layer was then deposited on the substrate by an atomic layer deposition (ALD) technique (trimethylaluminum and water as precursors, 150 °C), followed by opening a window by reactive ion etching (50 sccm CF₄, 5.0 Pa, 100 W, 10 min). Next, source and drain electrodes were formatted on the dielectric layer by the aforementioned method. The substrate was then coated with a monolayer of hexamethyldisilazane and immersed in the semiconducting CNT solution at 60 °C for 2 h. The substrate loaded with the CNT film was washed in toluene and isopropyl alcohol (IPA) for 5 min each. Finally, the CNT film was patterned by photolithography and oxygen plasma etching (180 sccm O₂, 200 W, 2 min) to form channels, and the solution of CsPbBr₃-QDs was spin-coated onto the top of the channels at 3000 rpm for 60 s, which avoids the

performance degradation of the CsPbBr$_3$-QDs caused by polar solvents (e.g., Remover PG and isopropyl alcohol) used in the fabrication processes.

**Fabrication of a sensor array on a flexible substrate.** A flexible sensor array was fabricated on a 125-μm-thick polyethylene naphthalate (PEN) substrate (Teijin DuPont Films) (Supplementary Fig. 27). The substrate was first heated at 190 °C for 3 h and washed by a solvent of Remover PG (Microchem) and IPA for 20 min each to remove particles on the substrate generated during the preheating. Next, the gate electrodes and interconnections (Ti/Au: 5/50 nm) were formatted by photolithography, EBV, and lift-off processes. Subsequently, an insulating 80-nm-thick Al$_2$O$_3$ layer was deposited on the substrate by an ALD technique, followed by opening a window by wet etching using phosphoric acid at 70 °C for 6 min. Finally, the source, drain electrodes, interconnections, and semiconducting channels were fabricated by the same processes using for the device on the rigid substrate.

**Characterization.** The materials and devices were characterized using an optical microscope (Nikon Eclipse LV100ND), an SEM (FEI Nova NanoSEM430, acceleration voltage of 1 kV), an AFM (Bruker Dimension Icon), and a UV–Vis–NIR spectroscope (Varian Cary 5000). The electrical and optoelectronic performances were measured using a semiconductor parameter analyzer (Agilent B1500A), a probe station (Cascade M150), an input signal generator (Tektronix AFG 3022C), an oscilloscope (Tektronix MSO 2024B), and a laser diode controller (Thorlabs ITC4001, using laser excitations of 405 and 516 nm) in a dark room at room temperature. The noise was measured by a noise measurement system (PDA NC300L, 100 kHz bandwidth). With the help of special mask to avoid crosstalk issues (Supplementary Fig. 30), the electrical performance of the 1024 phototransistors in the optoelectronic sensor array was automatically measured using a home-built transistor array test system (Agilent B1500A and Keysight 34980A) controlled by a self-developed program, and the data analysis and image processing were carried out using MATLAB (Supplementary Figs. 31–33).

**Statement of consent to publication of human face.** An image of a recognizable person in Fig. 3g and Supplementary Fig. 37c is the face of Qian-Bing Zhu who is the first author of this paper. The authors affirm that human research participants provided informed consent for publication of the images in Fig. 3g and Supplementary Fig. 37c.

## Data availability

The data that support the findings of this study are available at Zenodo (2021), https://doi.org/10.5281/zenodo.4540948.

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

## Acknowledgements

This work was supported by the National Key Research and Development Program of China (2020YFA0714702, 2016YFB041104), the National Natural Science Foundation of China (No. 61574143, 51532008, 61704175, 51502304, 22075312, 21773292, and 61874054), the Strategic Priority Research Program of Chinese Academy of Sciences (XDB30000000), the Key Research Program of Frontier Sciences of the Chinese Academy of Sciences (ZDBS-LY-JSC027, QYZDB-SSW-SLH031), Liaoning Revitalization Talents Program (XLYC1807109), the Thousand Talent Program for Young Outstanding Scientists, Key-Area Research and Development Program of Guangdong Province (2019B010934001), the Shandong Natural Science Foundation of China (ZR2019ZD49), and the projects supported by Shenyang National Laboratory for Materials Science, Institute of Metal Research, Chinese Academy of Sciences and State Key Laboratory of Luminescence and Applications, Chinese Academy of Sciences (L2019F28, Project Young Merit Scholars, SKLA-2019-03). The authors wish to thank Qin-Yi Zhang, Ya-Hui Li, Jin-Bo Wu, and Hua-Hua Li for valuable discussions. We thank Feng-Xiu Yuan for kind help on data processing.

## Author contributions

H.-M.C. and D.-M.S. conceived the idea and supervised the project. Q.-B.Z. and B.L. were equal major contributors to this work. Q.-B.Z. and B.L. performed the device fabrication. Q.-B.Z. carried out electrical and optoelectronic characterizations. C.L. proposed the mechanism of the device assisted by Q.-B.Z. Y.-N.T. carried out the simulation. Q.-B.Z. and B.L. contributed to the 3D schematic and illustration (Figs. 1a and 3d). Y.S. and B.L carried out the ALD depositions. Q.-B.Z., B.L., S.F., and M.-L.C. were responsible for characterizing the materials. X.S. carried out characterizations of noise power density supervised by X.-M.W. D.-D.Y. prepared the solution of CsPbBr₃-QDs supervised by X.-M.L. and H.-B.Z. S.Q. and Q.-W.L. prepared the semiconducting CNT solution. Q.-B.Z. and D.-M.S. wrote the paper. All authors discussed the results and commented on the manuscript.

## Competing interests

The authors declare no competing interests.
