## [Peer Review File · Nature Communications]

REVIEWER COMMENTS

Reviewer #1 (Remarks to the Author):

The manuscript "A flexible ultrasensitive optoelectronic sensor array for neuromorphic vision system" presents a study ranging from optoelectronic sensor device to neuromorphic vision systems. By using a perovskite/CNTs blend as the electro-photoactive layer, the transistor devices exhibit a high sensitivity to optical input, and can simultaneously complete signal sensing and processing tasks. Additionally, a flexible sensor array was prepared and employed for neuromorphic vision systems. The combination of carbon nanotubes and perovskite quantum dots contribute to the extraordinary sensitivity to light. The fascinating 32*32 integrated sensor array enables the demonstration of an efficient neuromorphic vision system based on this realistic sensor array. The manuscript is well written and the figures are clearly constructed. I recommend publication in Nature Communications with major revisions, the authors should consider the following comments:

1. In this study, the CsPbBr₃-QDs was used as photon absorption unit. While, in Figure 1c, according to the scale bar (2 μm), the particle size is around 100 nm, which is much larger than the size of quantum dots (typically 2-20 nm). Does QD aggregate? Additionally, are the CsPbBr₃ distributed on the surface, in the bulk, or at the bottom of CNTs? The distribution of CsPbBr₃ can obviously affect the optoelectronic properties. Please comment.
2. According to Supplementary Fig. 4, the introduction of CsPbBr₃ materials into CNT induce obvious device-to-device variations in the OFF state. While neuromorphic computation is considered to be resilient to hardware defects, device variability is costly. For the application of neuromorphic computation, high-density integration and mass production will not be possible until the variability. The evaluation of spatial and temporal reproducibility is essential, please comment.
3. Hysteresis behaviors of the transistors with the CNT channels and CNTs/QDs channels for the devices should be reported, because readers would be interested on the sweep rate dependence and bias stress effects.
4. It is noted that the sensor arrays with one-transistor configuration may induce the crosstalk issues between the neighboring sensor units since it will be uncertain state if the neighboring devices remains the ON state. However, the remarkable measured number "8" pattern has been obtained in Fig. 3e. The key role should be clearly explained why the selector transistor or diode were not included.
5. The measurement principle and sequence for the optoelectronic sensor array should be introduced to show how long it takes to achieve one measured pattern.
6. The high detectivity of the sensor with the combination channel is caused by the photogating effect, which is induced by the accumulation of electrons at the CNT-QD interface. Why is it necessary to consider the electron injection from the CNT into the QDs? What is the effect for the bandgap bending when the negatively-charged QDs induce positive carriers in the CNT film?
7. What is the result of the long-term photocurrent stability test, which would be very important for evaluating the possibility of practical application of such hybrid channels.

Reviewer #2 (Remarks to the Author):

The manuscript "A flexible ultrasensitive, ..." describes a flexible optoelectronic sensor array using a combination of carbon nanotubes and perovskite quantum dots as active channels, and the multifunction in terms of light responding, information storage and mimicking synaptic plasticity has been thoroughly studied. Overall, the key advances over previous reports include (1) phototransistor with a hybrid channel enables a high responsivity and a high specific detectivity, (2) sophisticated 1024 sensor array circuit experimentally demonstrates the functions for neuromorphic vision systems. I am certain that this manuscript will be of high interest to the readership of the journal and to the broader electronic community. This manuscript is well-organized and written, however, the authors must address the following comments:

1. It can be seen from the manuscript that the photocarriers of perovskite quantum dot film are only transferred between carbon nanotube layer and perovskite quantum dot layer, which means that the perovskite quantum dot film in the channel is not conductive. In order to prove that the mechanism is correct, the author should provide some representative data in terms of the

conductivity in the horizontal direction of the perovskite quantum dot film.

2. According to supplementary Figs. 9a and 14b, can we infer that the phototransistor can respond to a weaker light with a power density lower than 0.01 $\mu\text{W}/\text{cm}^2$? If so, it is better to give a response curve with weaker light to prove the authors' claim of ultrasensitive characters.
3. The perovskite quantum dot is a key material of photon absorption for the present phototransistors. Authors should perform more characterization of CsPbBr₃-QDs, including TEM and XRD, to show how special they are compared to other reported materials.
4. In order to consider the characteristics, the transfer and output curves of phototransistors are essential. Authors should give IDS-VDS characteristics of the device under light and dark conditions to extensively evaluate the phototransistor.
5. It can be seen from the Supplementary Fig. 11 that the rise (decay) time here is defined as the time for the photocurrent to rise (decay) from 10% of the final value to 90% (90 to 10%). The definition of response time should be mentioned for a better reader understanding.
6. Authors claimed the flexibility of the sensor array using the flexible materials of carbon nanotubes, however the dielectric layer used in this study is rigid Al₂O₃, it is well-known that if a large bending strain is applied, the leakage current may occur, therefore the author should provide more data on leakage currents about the optoelectronic performance in Supplementary Fig. 13.
7. Comparing the diagram in Fig. 1b with 3c, why the gate electrode design in a single device (inset of Fig. 1b) and the device in the array (Fig. 3c) are different, especially referring to the gap between source (drain) and gate electrodes in the sensor array.
8. In the top panel of Supplementary Fig. 16, I think lighting power density on the vertical y-axis corresponds to the time on the horizontal x-axis, but the values of the power density are denoted on the x-axis, please correct it to avoid misunderstanding for readers.

Reviewer #3 (Remarks to the Author):

In this manuscript, authors have provided the motivation for the development of artificial neuromorphic visual systems to simulate the flexibility, complexity and adaptability of biological vision systems. Here, they presented a flexible optoelectronic sensor array of 1024 pixels using a combination of carbon nanotubes and perovskite quantum dots as active materials for an efficient neuromorphic vision system. The device has an extraordinary sensitivity to light with a responsivity of 5.1×10^7 A/W and a specific detectivity of 2×10^{16} Jones, and demonstrates neuromorphic reinforcement learning by training the sensor array with a weak light pulse. The work is interesting in neuromorphic applications but it seems to be an extended work of published report by Jinxin Li et al, 2020. (DOI: 10.1002/aelm.202000535). Despite this there are several issues which hinder this work to be published in worthy journal "Nature Communications". The issues are as follows:

1. The "introduction" part of this manuscript lacks the importance and significance of CNTs/CsPbBr₃-QDs hybrid structure as efficient channel material.
2. The size and dimension of CNTs are missing. The description and nature of CNTs (single- or multi-wall) is important to discuss.
3. Authors claimed that CNTs are non-metallic as they did find any M11 peak in UV-Vis-NIR spectrum but only found S11 and S22 peaks which demonstrate the semiconducting behavior. So, this claim needs reference or further detail explanation.
4. Why absorption of CNTs is less as compared to CNTs/CsPbBr₃-QDs hybrid structure?
5. What is the energy band gap of CNTs if these are semiconductors?
6. In Figure 1e, the band diagram carry less information like values of band gap, electron affinity, etc....
7. The current on-off ratio in Figure 1d does not look like $\sim 10^7$ (in dark and under light).
8. It is point of explanation that at ~ 516 nm the absorption is meager but in Figure 1d the effect of light is significant.
9. Authors claimed that the major photo-generation is occurred at interface of CNTs and CsPbBr₃-QDs but no solid proof is given even transient PL has also lacks appropriate explanation.
10. It should be better if authors provide the I_{ds} - V_g curves for only CNTs FETs and CsPbBr₃-QDs FETs to distinguish the effect separately.
11. Please explain the mechanism of how effective splitting of exciton has been achieved when charges are transferred b/w CNTs and CsPbBr₃-QDs.

12. It is better to explain more about $\tau = \sum_i A_i \tau_i^2 / \sum_i A_i \tau_i$, and especially about the weight factor "A_i".
13. In Figure S8, what is the time period of light irradiation?
14. The Resp, EQE and Dect looks similar at 405 and 516 nm.
15. Authors should provide the time-dependent photocurrent of CNTs/ CsPbBr₃-QDs devices at different light power density and estimate the rise and decay time.
16. Can authors provide the responsivity at different V_g (-5, 0 and +5V) to analyze the recombination at Quasi-fermi levels with also proper band diagrams.
17. There is a confusion in decaying behavior that when gate pulse is given from 5V to 0V. So, at this situation why authors found rapid decay?
18. [IMPORTANT] I am not sure that the detectivity values provided by the authors are actually valid. There is a big miss-interpretation in the 2D community where the formula for shot-noise limited detectors is always used despite the fact that this might not be the case. See:
<https://www.nature.com/articles/s41566-018-0288-z>
<https://www.nature.com/articles/s41467-018-07643-7>
I urge the authors to address this point.
19. Figure 3, more explanation is required about when pulse numbers are increased the image become clearer.

Responses to the comments of reviewers

Reviewer #1:

The manuscript “A flexible ultrasensitive optoelectronic sensor array for neuromorphic vision system” presents a study ranging from optoelectronic sensor device to neuromorphic vision systems. By using a perovskite/CNTs blend as the electro-photoactive layer, the transistor devices exhibit a high sensitivity to optical input, and can simultaneously complete signal sensing and processing tasks. Additionally, a flexible sensor array was prepared and employed for neuromorphic vision systems. The combination of carbon nanotubes and perovskite quantum dots contribute to the extraordinary sensitivity to light. The fascinating 32*32 integrated sensor array enables the demonstration of an efficient neuromorphic vision system based on this realistic sensor array. The manuscript is well written and the figures are clearly constructed. I recommend publication in Nature Communications with major revisions, the authors should consider the following comments:

Response:

Thank you very much for your positive comments.

1. In this study, the CsPbBr₃-QDs was used as photon absorption unit. While, in Figure 1c, according to the scale bar (2 μm), the particle size is around 100 nm, which is much larger than the size of quantum dots (typically 2-20 nm). Does QD aggregate? Additional, are the CsPbBr₃ distributed on the surface, in the bulk, or at the bottom of CNTs? The distribution of CsPbBr₃ can obviously affect the optoelectronic properties.

Please comment.

Response:

We thank the reviewer for the valuable comments. In the original submission, the atomic force microscope (AFM) image of CsPbBr₃-QD film provided was not clear due to the large scanning range. Figure R1a shows the AFM image of the CsPbBr₃-QD film with a relatively smaller scanning range, indicating a small particle size. Figures R1b and R1c show the accurate size of CsPbBr₃-QD with a mean size of ~8.5 nm, which is characterized by the transmission electron microscope (TEM).

Figure R1 Characterization of CsPbBr₃-QD size. (a) AFM image of a CsPbBr₃-QD film (scale bar, 500 nm). (b) Histogram of the statistical distribution of the size of CsPbBr₃-QDs. (c) TEM image of CsPbBr₃-QDs (scale bar, 40 nm).

On the other hand, the CsPbBr₃-QD layer is placed on the top surface of CNT channel to ensure the formation of the high-quality CNT/CsPbBr₃-QD interface, that is, the deposition of the CsPbBr₃-QD layer is the last step in the device fabrication processes, which avoids the performance degradation of the CsPbBr₃-QDs caused by polar solvents (e.g. Remover PG and isopropyl alcohol) used in the fabrication processes. For more details, please refer to the Methods.

Corresponding modifications have been made on Page 15 (Lines 2-4) in blue, and in Figs. 1c in the revised manuscript as well as Supplementary Fig. 4 in the revised Supplementary Information.

2. According to Supplementary Fig. 4, the introduction of CsPbBr₃ materials into CNT induce obvious device-to-device variations in the OFF state. While neuromorphic computation is considered to be resilient to hardware defects, device variability is costly. For the application of neuromorphic computation, high-density integration and mass production will not be possible until the variability. The evaluation of spatial and temporal reproducibility is essential, please comment.

Response:

We thank the reviewer for the valuable comments. The introduction of CsPbBr₃-QDs into the CNT channel really induced a current fluctuation of about one order of magnitude in the OFF state to induce device-to-device variations. On the other hand, the current in the ON state is uniform, and the ON/ OFF ratio is greater than 10⁵, which indicates that the signal-to-noise ratio is sufficiently high for neuromorphic

computation. We measured the transfer characteristics of 1024 devices in the sensor array under dark condition, as shown in Fig. R2, which demonstrates the excellent performance uniformity, spatial and temporal reproducibility of the device.

Figure R2 Statistical analysis of the devices in the sensor array. (a) Typical transfer characteristics of 1024 pixels were measured under dark conditions showing uniform electrical performance at $V_{DS} = 1$ V. (b) Statistical distribution of the I_{ON} with a variance of 13.9%. (c) Statistical distribution of the I_{OFF} fitted with a logarithmic normal curve with a variance of 3.1%. (d) Statistical distribution of the on/off ratio fitted with a logarithmic normal curve with a variance of 5.6%.

3. Hysteresis behaviors of the transistors with the CNT channels and CNTs/QDs channels for the devices should be reported, because readers would be interested on the sweep rate dependence and bias stress effects.

Response:

Thanks for your suggestions. Figure R3 shows the transfer characteristics of 12 transistors with CNT channel and CNTs/CsPbBr₃-QD channel under ambient dark conditions. The sweep swing of V_{GS} is from -5 V to 5 V, and the sweep time is 30 s.

Figure R3 Transfer characteristics of 12 transistors with CNT channel and CNTs/CsPbBr₃-QD channel under ambient dark conditions. $V_{DS} = 1$ V.

Corresponding modifications have been made in Supplementary Fig. 11 in the revised Supplementary Information.

4. It is noted that the sensor arrays with one-transistor configuration may induce the crosstalk issues between the neighboring sensor units since it will be uncertain state if the neighboring devices remains the ON state. However, the remarkable measured number “8” pattern has been obtained in Fig. 3e. The key role should be clearly explained why the selector transistor or diode were not included.

Response:

We thank the reviewer for the valuable comment. The sensor array with one-transistor configuration may lead to crosstalk issue between neighboring sensor units if the neighboring devices remain the ON state. In order to measure number “8” pattern,

we designed a special mask shown in Fig. R4a (the devices in the white region are ON state). If the devices in red region remain OFF state (dark condition), the blue area can avoid crosstalk issues. For an instance, the device A is measuring phototransistor which remains OFF state, and the device B and D are neighboring phototransistors which remain the ON state (Fig. R4b). The green line represents the actual current value of device A, and the red line represents the crosstalk current flowing through devices B, C and D. In our design, we designed a special mask to keep device C in OFF state to avoid crosstalk issues. In order to achieve more complex images in future, such as face images with finer and more complex features, and super-resolution for face images is more challenging without the selector transistor or diode.

Figure R4 Designed principle avoiding crosstalk issue.

Corresponding modifications have been made on Page 16 (Lines 3-4) in blue in the revised manuscript as well as Supplementary Fig. 30 in the revised Supplementary Information.

5. The measurement principle and sequence for the optoelectronic sensor array should

be introduced to show how long it takes to achieve one measured pattern.

Response:

We thank the reviewer for the comments. The measurement process is introduced as follows. The special mask was put on the sensor array and the laser is turned on (Supplementary Fig. 30). Next, all 1024 devices were switched off by applied $V_{GS} = 5$ V to the 32 gate channel multiplexers. Finally, the I_{DS} of each pixel in the sensor array were measured by sequentially connecting or disconnecting the circuit through the multiplexer. The time required to achieve a measured pattern depends on the time to measure I_{DS} of one pixel. For example, in order to show an ordinary pattern, it takes less than 1 s for one pixel, and it takes about 1000 s in total.

Corresponding modifications have been made in Supplementary Fig. 32 in the revised Supplementary Information.

6. The high detectivity of the sensor with the combination channel is caused by the photogating effect, which is induced by the accumulation of electrons at the CNT-QD interface. Why is it necessary to consider the electron injection from the CNT into the QDs? What is the effect for the bandgap bending when the negatively-charged QDs induce positive carriers in the CNT film?

Response:

We thank the reviewer for the valuable comments. Figure R5 shows the energy band diagram of the photogating mechanism at the CNT/CsPbBr₃-QD interface. The built-in electric field from QDs to CNTs shown in the top panel leads to a band bending

at the interface under dark condition. For the light-on state on the bottom panel, the photo-generated electron-hole pairs can be effectively separated and transferred to the CNTs in the built-in electric field, but the photo-generated electrons are trapped in the QDs. The direction of electric field generated by the trapped electrons in the QDs is opposite to the built-in field, so the degree of bandgap bending can be reduced.

Figure R5 Energy band diagrams at the light-off state (top panel) and light-on state (bottom panel).

Corresponding modifications have been made in Supplementary Fig. 15 in the revised Supplementary Information.

7. What is the result of the long-term photocurrent stability test, which would be very important for evaluating the possibility of practical application of such hybrid channels.

Response:

We thank the reviewer for the valuable comments. Figure R6 shows the stability of the phototransistor under ambient condition. The transfer curves under dark and illuminated conditions (0.9 μW/cm²) were recorded for 8 months. Based on the excellent stability of CNTs and CsPbBr₃-QDs, the maintained optoelectronic characteristics indicate the long-term stability of photocurrent.

Figure R6 Stability of the electrical and optoelectronic performance of CsPbBr₃-QD transistor at room temperature. $V_{DS} = 1$ V, $\lambda = 405$ nm.

Corresponding modifications have been made on Page 7 (Lines 17-19) in blue in the revised manuscript as well as Supplementary Fig. 21 in the revised Supplementary Information.

Reviewer #2:

The manuscript “A flexible ultrasensitive, ...” describes a flexible optoelectronic sensor array using a combination of carbon nanotubes and perovskite quantum dots as active channels, and the multifunction in terms of light responding, information storage and mimicking synaptic plasticity has been thoroughly studied. Overall, the key advances over previous reports include (1) phototransistor with a hybrid channel enables a high responsivity and a high specific detectivity, (2) sophisticated 1024 sensor array circuit experimentally demonstrates the functions for neuromorphic vision systems. I am certain that this manuscript will be of high interest to the readership of the journal and to the broader electronic community. This manuscript is well-organized and written, however, the authors must address the following comments:

Response:

Thank you very much for your positive comments.

1. It can be seen from the manuscript that the photocarriers of perovskite quantum dot film are only transferred between carbon nanotube layer and perovskite quantum dot layer, which means that the perovskite quantum dot film in the channel is not conductive. In order to prove that the mechanism is correct, the author should provide some representative data in terms of the conductivity in the horizontal direction of the perovskite quantum dot film.

Response:

Figure R7 shows the transfer ($I_{DS}-V_{GS}$) characteristics of the CsPbBr₃-QD

transistor under dark and illuminated conditions. The sub-picoampere current close to the noise level indicates that the CsPbBr₃-QD film is not conductive in the horizontal direction.

Figure R7 I_{DS} - V_{GS} characteristics of CsPbBr₃-QD transistor. $V_{DS} = 1$ V, $\lambda = 405$ nm.

Corresponding modifications have been made on Page 4 (Lines 19-20) in blue in the revised manuscript as well as Supplementary Fig. 8 in the revised Supplementary Information.

2. According to supplementary Figs. 9a and 14b, can we infer that the phototransistor can respond to a weaker light with a power density lower than 0.01 $\mu\text{W}/\text{cm}^2$? If so, it is better to give a response curve with weaker light to prove the authors' claim of ultrasensitive characters.

Response:

We try our best to carry out the measurement under extra-weak lighting environment. Figure R8 shows that the photodetector can respond to a weaker light with a lighting power density lower than 0.01 $\mu\text{W}/\text{cm}^2$, which is the limited resolution of our current measurement equipment.

Figure R8 Light-dose-dependent response of the phototransistor at various lighting power densities. The device not only exhibits a time-dependent response, but also light-intensity-dependent characteristics. $V_{DS} = 1$ V, $V_{GS} = 5$ V, $\lambda = 405$ nm.

Corresponding modifications have been made in Supplementary Fig. 13 in the revised Supplementary Information.

3. The perovskite quantum dot is a key material of photon absorption for the present phototransistors. Authors should perform more characterization of CsPbBr₃-QDs, including TEM and XRD, to show how special they are compared to other reported materials.

Response:

Thanks for this valuable suggestion. Figure R9 shows the TEM and XRD characterizations of the CsPbBr₃-QDs. The CsPbBr₃-QDs exhibit a cubic crystal structure with a mean size of ~8.5 nm. To guarantee the electrical conductivity, the QDs should be purified enough to exclude the excessive organic components. For conventional synthesis strategy (with oleic acid and oleylamine as ligands), the trap states will form after several purification cycles, inducing a reduced quantum yield. As demonstrated in Fig. 1e, the QDs should possess some trap states to trap electrons.

Therefore, it is not necessary to make the quantum yield high in order to make the existence of trap states more favorable. The quantum yield and photoluminescence decay measurements accord well with the existence of trap states. The quantum yield of QDs is about 85% and the photoluminescence decay is not mono-exponential as shown in supplementary Fig. 12b.

Figure R9 Characterization of CsPbBr₃-QDs. (a) TEM image of CsPbBr₃-QDs (scale bar, 40 nm). (b) Histogram of statistical distribution of the size of CsPbBr₃-QDs. (c) XRD characteristics.

Corresponding modifications have been made in Supplementary Fig. 4 in the revised Supplementary Information.

4. In order to consider the characteristics, the transfer and output curves of

phototransistors are essential. Authors should give I_{DS} - V_{DS} characteristics of the device under light and dark conditions to extensively evaluate the phototransistor.

Response:

Figure R10 shows the I_{DS} - V_{DS} characteristics at various lighting power densities for fixed $V_{GS} = 5\text{ V}$ and 0 V , indicating that the optoelectronic performance of the phototransistor is related to the lighting power density. Even when the lighting power density is less than $0.01\text{ }\mu\text{W}/\text{cm}^2$, the device has an obvious response, and shows similar trends when using different V_{GS} .

Figure R10 I_{DS} - V_{DS} curves under various lighting power densities with a fixed V_{GS} . (a) $V_{GS} = 5\text{ V}$. (b) $V_{GS} = 0\text{ V}$. $\lambda = 405\text{ nm}$.

Corresponding modifications have been made on Page 4 (Lines 20-22) and Page 5 (Line 1) in blue in the revised manuscript as well as Supplementary Fig. 9 in the revised Supplementary Information.

5. It can be seen from the Supplementary Fig. 11 that the rise (decay) time here is defined as the time for the photocurrent to rise (decay) from 10% of the final value to 90% (90 to 10%). The definition of response time should be mentioned for a better

reader understanding.

Response:

According to your suggestions, we have added the definition of response time in the revised manuscript.

Corresponding modifications have been made in Supplementary Fig. 19 in the revised Supplementary Information.

6. Authors claimed the flexibility of the sensor array using the flexible materials of carbon nanotubes, however the dielectric layer used in this study is rigid Al_2O_3 , it is well-known that if a large bending strain is applied, the leakage current may occur, therefore the author should provide more data on leakage currents about the optoelectronic performance in Supplementary Fig. 13.

Response:

Figure R11 shows the $I_{\text{DS}}-V_{\text{GS}}$ and $I_{\text{GS}}-V_{\text{GS}}$ characteristics of a flexible device with a given bending strain ($\varepsilon = 0.4\%$) under dark and illuminated conditions ($P = 1.43 \mu\text{W}/\text{cm}^2$), and the sub-picoampere leakage current demonstrates the robustness of the device.

Figure R11 I_{DS} - V_{GS} and I_{GS} - V_{GS} characteristics of the flexible device given a fixed bending strain ($\varepsilon = 0.4\%$) under dark and illuminated conditions ($P = 1.43 \mu\text{W}/\text{cm}^2$). $V_{DS} = 1 \text{ V}$, $\lambda = 405 \text{ nm}$.

Corresponding modifications have been made in Supplementary Fig. 23 in the revised Supplementary Information.

7. Comparing the diagram in Fig. 1b with 3c, why the gate electrode design in a single device (inset of Fig. 1b) and the device in the array (Fig. 3c) are different, especially referring to the gap between source (drain) and gate electrodes in the sensor array.

Response:

The probability of leakages of the devices in the sensor array increases exponentially as the scale of array chips continues to increase. The leakage suppression is crucial to improve the success rate of the array fabrication. Therefore, we optimized the gate structure of our phototransistors by reducing the overlap area between the gate and the source/drain electrodes (as shown in Fig. 3c), thereby significantly increasing the yield to 100% (Supplementary Fig. 29).

8. In the top panel of Supplementary Fig. 16, I think lighting power density on the vertical y-axis corresponds to the time on the horizontal x-axis, but the values of the power density are denoted on the x-axis, please correct it to avoid misunderstanding for readers.

Response:

Thanks for your question. The previous statement in the original submission was

unclear, which caused readers to misunderstand. According to the reviewer's suggestion, we corrected the positions of the power density.

Figure R12 Light-intensity-dependent synaptic plasticity with a pulse width of 20 ms. $V_{DS} = 1$ V, $\lambda = 516$ nm.

Corresponding modifications have been made in Supplementary Fig. 26 in the revised Supplementary Information.

Reviewer #3:

In this manuscript, authors have provided the motivation for the development of artificial neuromorphic visual systems to simulate the flexibility, complexity and adaptability of biological vision systems. Here, they presented a flexible optoelectronic sensor array of 1024 pixels using a combination of carbon nanotubes and perovskite quantum dots as active materials for an efficient neuromorphic vision system. The device has an extraordinary sensitivity to light with a responsivity of 5.1×10^7 A/W and a specific detectivity of 2×10^{16} Jones, and demonstrates neuromorphic reinforcement learning by training the sensor array with a weak light pulse. The work is interesting in neuromorphic applications but it seems to be an extended work of published report by Jinxin Li et al, 2020. (DOI: 10.1002/aelm.202000535). Despite this there are several issues which hinder this work to be published in worthy journal “Nature Communications”. The issues are as follows:

Response:

Thank you very much for your positive comment.

As pointed out by the reviewer, an optoelectronic synapse has been reported with a two-terminal device structure and a hybrid material of multi-walled-CNTs and organic-inorganic MAPbBr₃-QDs by Li et al, 2020. However, in our study, we have demonstrated a phototransistor with a combination of semiconducting-CNTs and all-inorganic CsPbBr₃-QDs as an active material. It is worth mentioning that the materials and device structures are significantly different from the previous report, resulting in a completely different work mechanism and excellent optoelectronic performance.

Specifically, for neuromorphic optoelectronic computing, optoelectronic sensors with ultra-high responsivity, good detectivity, high signal-to-noise ratio and long-term stability are required. Compared with multi-walled-CNTs, semiconducting-CNTs can significantly improve the signal-to-noise ratio of the lighting response due to their excellent on/off ratio. Also, all-inorganic CsPbBr₃-QDs are more stable than organic-inorganic MAPbBr₃-QDs. Fundamentally, the semiconducting-CNT/CsPbBr₃-QD phototransistor provides a photogating effect, which can achieve an ultrahigh light to dark current ratio ($>10^6$) and light response at a low lighting power density ($<0.01 \mu\text{W}/\text{cm}^2$) to obtain an extraordinary sensitivity to light with a responsivity of $5.1 \times 10^7 \text{ A/W}$ and a specific detectivity of 2×10^{16} Jones, and further demonstrates neuromorphic reinforcement learning by training the sensor array with a weak light pulse of $1 \mu\text{W}/\text{cm}^2$. In sharp contrast, the multi-walled-CNT/MAPbBr₃-QD optoelectronic synapse is a two-terminal device and the optoelectronic mechanism is the photoconductive effect, which leads to a low light to dark current ratio (< 2) under a lighting power density of $125 \mu\text{W}/\text{cm}^2$. Obviously, our optoelectronic performance of the semiconducting-CNT/CsPbBr₃-QD phototransistor is greatly better than that of the multi-walled-CNT/MAPbBr₃-QD optoelectronic synapse, and the long-term stability shown in our device (Fig. R6) originates from the use of all-inorganic CsPbBr₃-QD materials.

Therefore, we believe that our work will enlighten the community on the development of neuromorphic vision systems and should be regarded as a major progress in this field rather than an incremental work.

1. The “introduction” part of this manuscript lacks the importance and significance of CNTs/CsPbBr₃-QDs hybrid structure as efficient channel material.

Response:

Thanks for your valuable comments. For the development of a high-performance neuromorphic vision system, optoelectronic sensors with ultra-high responsivity, detectivity and signal-to-noise ratio are necessary to offer enhanced imaging capability under extreme dim light conditions. For the selection of active sensing materials, the all-inorganic perovskite CsPbBr₃-QDs have excellent optoelectronic response performance, and CNTs can significantly improve the detection signal-to-noise ratio of the sensor due to the excellent carrier mobility and on/off ratio. Both materials can be fabricated into uniform large-area films with excellent flexibility and stability, and the combination of these two materials provides a new strategy for the design and fabrication of high-performance neuromorphic vision sensors. We have added the importance and significance of CNT/CsPbBr₃-QD hybrid structure as efficient channel material in the introduction part of the revised manuscript.

Corresponding modifications have been made on Page 3 (Lines 6-15) in blue in the revised manuscript.

2. The size and dimension of CNTs are missing. The description and nature of CNTs (single- or multi-wall) is important to discuss.

Response:

The raw arc-discharge CNT purchased from Carbon Solution Inc.

(<https://carbonsolution.com/products/ap-swnt>) has a narrow diameter distribution with a peak value of 1.55 ± 0.1 nm and a bundle length of 1-5 μm . Furthermore, high-purity (>99.9%) semiconducting-CNTs were sorted by mixing these bulk CNTs in a xylene solution, followed by ultrasonic stirring, and then centrifuging to remove CNT bundles and insoluble substances. We measured the CNT length based on a low-density CNT film on a SiO_2/Si substrate. Figure R13 shows the morphology of the sparse CNT network and a mean length of the CNTs is 1.4 μm based on the statistical analysis of the length distribution.

Figure R13 Morphology and CNT length distribution of semiconducting-CNT film. (a) SEM image of semiconducting-CNT film with a deposition time of 30 minutes (scale bar, 1 μm). (b) Histogram of statistical distribution of the length of semiconducting-CNTs.

Corresponding modifications have been made on Page 13 (Lines 17-20) in blue in the revised manuscript as well as Supplementary Fig. 3 in the revised Supplementary Information.

3. Authors claimed that CNTs are non-metallic as they did find any M11 peak in UV-Vis-NIR spectrum but only found S11 and S22 peaks which demonstrate the

semiconducting behavior. So, this claim needs reference or further detail explanation.

Response:

The purity of semiconducting-CNTs is evaluated from the absorption spectrum (Fig. R14) on a reported method (J. Ding, et al., *Nanoscale* 6, 2328, 2014). This method is applicable for the removal of the M₁₁ peak due to the elimination of metallic CNTs by calculated absorption peak ratio (ϕ_i). $\phi_i = A_{\text{CNT}}/(A_{\text{CNT}} + A_{\text{B}}) = 0.420$, where A_{CNT} (red area) is the enveloping area of the M₁₁ and S₂₂ bands enclosed by the linear baseline (dot line) corresponding to the amount of metallic- and semiconducting-CNTs in the sample, and A_{B} is the area covered by the linear baseline of the same region, mainly attributing to the amorphous carbon impurity. The purity of semiconducting-CNTs in our work exceeds 99.9%, because ϕ_i in our work is larger than that (0.404) in previous work reported by J. Ding et al.

Figure R14 Absorption spectrum of the purified semiconducting-CNTs for calculating absorption peak ratio (ϕ_i).

It should be noted that the spectral-based evaluation methods are not accurate enough to characterize the semiconducting-CNTs with a purity higher than 99%. Through electrical measurements of short-channel transistors, the purity of semiconducting-CNTs sorted using the same homopolymer (PCz) can reach more than

99.9% (J. Gu, et al., *Small*, 12, 4993, 2016; L. Liu, et al., *Science*, 368, 850, 2020). In our study, the transfer characteristics of 1024 transistors in the sensor array also proved the high purity of semiconducting-CNTs (Supplementary Fig. 29).

Corresponding modifications have been made in Supplementary Fig. 2 in the revised Supplementary Information.

4. Why absorption of CNTs is less as compared to CNTs/CsPbBr₃-QDs hybrid structure?

Response:

As pointed out by the reviewer, the CNTs have less absorption compared to the CNT/CsPbBr₃-QD hybrid structure, as shown in Fig. R15a. Because the thickness of the CNT channel is only 2-5 nm, the absorption of CNT film in the region of 300-900 nm is relatively weak. The all-inorganic perovskite CsPbBr₃-QD film can easily cover the absorption of the CNTs due to its excellent light absorption properties. Note that there is no obvious change between the hybrid film and the CsPbBr₃-QD film. As shown in Fig. R15b, the absorption can be enhanced when a thick CNT film (50 nm) was used.

Figure R15 Absorption spectrum. (a) Absorption spectra of the CNT film, CsPbBr₃-QD film and CNT/CsPbBr₃-QD film on a quartz substrate. (b) Absorption spectrum of a thick CNT film.

Corresponding modifications have been made in Supplementary Fig. 6 in the revised Supplementary Information.

5. What is the energy band gap of CNTs if these are semiconductors?

Response:

According to the absorption spectrum of the purified CNTs (Supplementary Fig. 1), the S₂₂ peak can be determined to be ~1000 nm, which can be inferred from the empirical Kataura plot (R. B. Weisman, et al., Nano Lett., 3, 1235, 2003) that the band gap of the semiconducting-CNTs is ~1.3 eV. Meanwhile, we can conclude that the valance band is about 5.1 eV (A. Javey, et al., Nature, 424, 654, 2003) and the conduction band can be inferred as 3.8 eV.

Corresponding modifications have been made in Supplementary Fig. 10 in the revised Supplementary Information.

6. In Figure 1e, the band diagram carry less information like values of band gap, electron affinity, etc....

Response:

Thanks for your valuable comments. Figure R16a shows a revised version of the energy band diagram, which includes more information including the work function of CNT of ~4.8 eV (S. Suzuki, et al, Appl. Phys. Lett., 76, 4007, 2000) and the band

structure of CsPbBr₃-QD (M. Xiao, et al, Adv. Funct. Mater., 29, 1905683, 2019).

Figure R16 Band gap diagram. (a) Band structure of CNT and CsPbBr₃-QD. (b) Energy band diagram under dark (top panel) and illuminated (bottom panel) conditions.

Corresponding modifications have been made in Fig. 1e in the revised manuscript as well as Supplementary Fig. 10 in the revised Supplementary Information.

7. The current on-off ratio in Figure 1d does not look like $\sim 10^7$ (in dark and under light).

Response:

We fully agree with your valuable comments. Indeed, the accurate value of the on- and off-current ratio is re-evaluated to be 3.6×10^6 , which is slightly less than 10^7 , as shown in Fig. R17.

Figure R17 I_{DS}-V_{GS} curves of the CNT/CsPbBr₃-QD phototransistor under dark and

illuminated conditions ($P = 17.4 \text{ mW/cm}^2$). $V_{DS} = 1 \text{ V}$, $\lambda = 506 \text{ nm}$.

Corresponding modifications have been made on Page 4 (Lines 16-19) in blue in the revised manuscript as well as Supplementary Fig. 7 in the revised Supplementary Information.

8. It is point of explanation that at $\sim 516 \text{ nm}$ the absorption is meager but in Figure 1d the effect of light is significant.

Response:

As shown in Fig. R18a, the absorption spectrum of the $\text{CsPbBr}_3\text{-QD}$ film exhibits a sharp absorption edge at 525 nm , indicating that the energy of the 516 nm photon is sufficient to excite electrons from the valence band to the conduction band. Therefore, it is reasonable that the device exhibited a significant response to the optoelectronic performance at $\sim 516 \text{ nm}$. Meanwhile, we measured the optoelectronic response to the wavelength of 638 nm , as shown in Fig. R18b, confirming that the $\text{CsPbBr}_3\text{-QD}$ is not sensitive to the wavelengths greater than the absorption edge.

Figure R18 Optoelectronic properties of $\text{CsPbBr}_3\text{-QDs}$ under different wavelength lasers. (a) Absorption spectrum of $\text{CsPbBr}_3\text{-QD}$ film on the quartz substrate. (b) I_{DS} - V_{GS} characteristics of the CNT/ $\text{CsPbBr}_3\text{-QD}$ phototransistor under dark and illuminated

conditions ($P = 53 \mu\text{W}/\text{cm}^2$). $V_{\text{DS}} = 1 \text{ V}$, $\lambda = 638 \text{ nm}$.

9. Authors claimed that the major photo-generation is occurred at interface of CNTs and CsPbBr₃-QDs but no solid proof is given even transient PL has also lacks appropriate explanation.

Response:

Thanks for your valuable comments. The CsPbBr₃-QDs are used as active materials that absorb photons to generate electron-hole pairs, and then a highly effective dissociation of photo-generated electron-hole pairs occur at the CNT/CsPbBr₃-QD interface. This is verified by the steady-state and transient PL spectra CNT, CsPbBr₃-QD and CNT/CsPbBr₃-QD films. The lifetime of excitons in PL measurement is detected at fixed excitation and emission wavelengths. Although the lifetime can be obtained by multi-exponential function fitting, all lifetimes correspond to radiative recombination with the same excited state. Different lifetime values are related to different recombination processes including exciton direct recombination and exciton trap/de-trap processes. Generally, fast decay can be considered to be related to direct recombination, while slow decay is related to recombination in other processes. Figure R19 shows that a highly effective splitting of photo-generated electron-hole pairs occurs at the CNT/CsPbBr₃-QD interface. The longer the exciton lifetime of the CsPbBr₃-QDs, the stronger its fluorescence intensity. Therefore, the exciton separation at the CNT/CsPbBr₃-QD interface is faster than that of the CsPbBr₃-QDs, which is related to the shorter exciton lifetime.

Figure R19 Transient PL spectra of CsPbBr₃-QD and CNT/CsPbBr₃-QD films.

Corresponding modifications have been made on Page 5 (Lines 10-13) in blue in the revised manuscript as well as Supplementary Fig. 12 in the revised Supplementary Information.

10. It should be better if authors provide the I_{DS} - V_{GS} curves for only CNTs FETs and CsPbBr₃-QDs FETs to distinguish the effect separately.

Response:

According to the valuable suggestions of the reviewer, we used CNTs and CsPbBr₃-QDs as the active channel to fabricate two types of field-effect transistors, namely CNT transistor and CsPbBr₃-QD transistor. The electrical and optoelectronic performances of the two types of transistors are compared to distinguish the effect separately (Fig. R20). The I_{DS} - V_{GS} characteristics of CNT transistor show normal electrical performance with an on- and off-current ratio of 10^6 , but there is almost no optical response for a laser with a wavelength of 405 nm and a power density of 173 $\mu\text{W}/\text{cm}^2$ (Fig. R20a). On the other hand, the inherent carrier mobility of CsPbBr₃-QDs is very low, and there is a large contact resistance between adjacent CsPbBr₃-QDs, thus

the CsPbBr₃-QD film exhibits non-conductive behavior (Fig. R20b). Therefore, channel materials composed only of CsPbBr₃-QD or CNT cannot meet the high optical response requirements.

Figure R20 Electrical and optoelectronic performances of CNT transistor and CsPbBr₃-QD transistor. (a) I_{DS} - V_{GS} characteristics of CNT transistor under dark and illuminated conditions ($P = 173 \mu\text{W}/\text{cm}^2$). (b) I_{DS} - V_{GS} characteristics of CsPbBr₃-QD transistor under dark and illuminated conditions ($P = 173 \mu\text{W}/\text{cm}^2$). $V_{DS} = 1 \text{ V}$, $\lambda = 405 \text{ nm}$.

Corresponding modifications have been made on Page 4 (Lines 19-20) in blue in the revised manuscript as well as Supplementary Fig. 8 in the revised Supplementary Information.

11. Please explain the mechanism of how effective splitting of exciton has been achieved when charges are transferred b/w CNTs and CsPbBr₃-QDs.

Response:

Thanks for your valuable comments. When the CsPbBr₃-QDs was spin-coated on the CNT channel, a built-in electric field was formed at the CNT/CsPbBr₃-QD interface, which is the key to effectively splitting excitons (Y. Liu, et al., Nat. Commun, 6, 8589, 2015). Figure R21 shows the negative shift of the I_{DS} - V_{GS} characteristics compared the

transistors with CNT channel and CNT/CsPbBr₃-QD channel under dark conditions, indicating the formation of a built-in electric field. The energy band diagrams of CNT and CsPbBr₃-QD show that electrons flow from CsPbBr₃-QD to CNT due to the difference in Fermi level, thus forming a space charge region and a built-in electric field (Fig. R17). This process can also be verified by the transient PL spectra of CsPbBr₃-QD and CNT/CsPbBr₃-QD films (Fig. R 19).

Figure R21 I_{DS} - V_{GS} characteristics of the transistors with CNT channel and CNT/CsPbBr₃-QD channel under dark condition.

Corresponding modifications have been made in Supplementary Figs. 11 and 12 in the revised Supplementary Information.

12. It is better to explain more about $\tau = \sum_i A_i \tau_i^2 / \sum_i A_i \tau_i$, and especially about the weight factor “ A_i ”.

Response:

Figure R19 shows that exciton dissociation at the CNT/CsPbBr₃-QD interface contributes to reduce exciton recombination, thereby shortening the lifetime. The weight factor “ A_i ” is derived from the lifetime decay fitting, as a mathematical operation, it represents the ratio of related lifetime “ τ_i ”. Considering that the

classification of multi-lifetime remains challenging, the average lifetime is usually used to compare the exciton populations, that is, the longer the lifetime, the higher exciton recombination density.

Corresponding modifications have been made in Supplementary Fig. 12 in the revised Supplementary Information.

13. In Figure S8, what is the time period of light irradiation?

Response:

The light-dose-dependent response includes time dependence and light intensity dependence. For the time-dependent characteristics, as the light irradiation time is extended, the photocurrent of the phototransistor increases and reaches saturation after a sufficient illumination duration. These phenomena can also be observed in the literature (F. Zhou, et al, Nat. Nanotechnol., 14, 776-782, 2019; X. Liu, et al, Nat. commun., 5, 4007, 2014). Figure R22 shows the time-dependent characteristics of the phototransistor at various lighting power densities during an illumination duration of 275 s, which indicates that the I_{DS} will reach a saturation value as the illumination continues.

Figure R22 Light-dose-dependent response of the phototransistor at various lighting power densities. The device not only exhibits a time-dependent response, but also light-intensity-dependent characteristics. $V_{DS} = 1$ V, $V_{GS} = 5$ V, $\lambda = 405$ nm.

Corresponding modifications have been made in Supplementary Fig. 13 in the revised Supplementary Information.

14. The Resp, EQE and Dect looks similar at 405 and 516 nm.

Response:

As the reviewer pointed out, for the different laser wavelengths of 405 and 516 nm, since CsPbBr₃-QD is highly sensitive to the light of 405 and 516 nm wavelengths, the phototransistor does show similar optoelectronic performance. Figure R23 shows a comparison of the responsivity, EQE and detectivity for the light of 405 and 516 nm wavelengths. This phenomenon can also be found in recently published literature (S. Lukman, et al., Nat. Nanotechnol., 15, 675, 2020).

Figure R23 Comparison of optoelectronic performances at the wavelengths of 405 and 516 nm. (a) Dependence of responsivity on P . (b) Dependence of EQE on P . (c) Dependence of detectivity on P . $V_{\text{DS}} = 1 \text{ V}$, $V_{\text{GS}} = 5 \text{ V}$.

15. Authors should provide the time-dependent photocurrent of CNTs/ CsPbBr₃-QDs devices at different light power density and estimate the rise and decay time.

Response:

Thanks for your valuable comments. Figure R24a shows the time-dependent photocurrents of CNT/CsPbBr₃-QD devices at different lighting power densities. The less time is required to reach photocurrent saturation with higher lighting power density. The dependence of the rise time and decay time on the lighting power density is shown in Fig. R24b. The rise time can decrease to 3.3 ms at $0.78 \text{ W}/\text{cm}^2$, and the decay time

is about 1 ms which is not associated with lighting power density.

Figure R24 Dependence of the response time on the lighting power density. (a) Light-dose-dependent response of the optoelectronic transistor at various lighting power densities (P). (b) Dependence of the rise and decay time on P . $V_{DS} = 1 \text{ V}$, $V_{GS} = 5 \text{ V}$, $\lambda = 516 \text{ nm}$.

Corresponding modifications have been made in Supplementary Fig. 20 in the revised Supplementary Information.

16. Can authors provide the responsivity at different V_g (-5, 0 and +5V) to analyze the recombination at Quasi-fermi levels with also proper band diagrams.

Response:

Thank for your valuable comments. According to the reviewer's suggestion, we measured the dependence of the photocurrent on the lighting power density (P) under various V_{GS} of -5 V , 0 V and $+5 \text{ V}$ (Fig. R25a), and further compared the variation of the responsivity. It can be found that the photocurrent and responsivity are very similar, especially in the cases of higher lighting power densities, which indicates that the negatively-charged QDs is the dominant factor leading to the increase in current. Therefore, the photocurrent mainly depends on the lighting power density and has no

obvious relationship with the gate voltage.

Figure R25c shows a built-in electric field that induces a band bending at the CNT/CsPbBr₃-QD interface and reaches the Fermi level equilibrium at different V_{GS} (-5, 0 and +5 V) under dark conditions. The built-in electric field is the strongest for $V_{GS} = -5$ V, because the direction of the electric field of $V_{GS} = -5$ V is consistent with the built-in electric field, but for $V_{GS} = 5$ V, the situation is reversed. Figure R25d shows the energy band diagram under illuminated conditions. The electrons are trapped in the QDs resulting in a photogating effect, which modulates the CNT into ON-state to increase the current. The photocurrent depends on the number of electrons trapped in the QDs, therefore, there is no obvious difference in the energy band diagram under different V_{GS} conditions.

Figure R25 Analysis of optoelectronic characteristics with various gate voltages. (a) Dependence of the photocurrent on P . (b) Dependence of responsivity on P . (c) Energy band diagram under dark condition at different V_{GS} . (d) Energy band diagram under

illuminated condition.

Corresponding modifications have been made on Page 6 (Lines 15-16) and Page 7 (Lines 1-3) in blue in the revised manuscript as well as Supplementary Fig. 15 in the revised Supplementary Information.

17. There is a confusion in decaying behavior that when gate pulse is given from 5V to 0V. So, at this situation why authors found rapid decay?

Response:

Thanks for your valuable question. We understand that the reviewer is concerned that the decay time should include the period of the gate pulse, and we fully agree with your valuable comments. More strictly, the decay time mentioned in the original submission should be the period calculated after the completion of the electrical reset pulse. Such operation method is usually used to improve the decay time of phototransistors in the literature (G. Konstantatos, et al., Nat. Nanotechnol., 7, 363-368, 2012; S. Jeon, et al., Nat. Mater., 11, 301-305, 2012; O. Lopez-Sanchez, et al., Nat. Nanotechnol., 8, 497-501, 2013). The fabricated phototransistor has image sensing function and also shows memory and synaptic characteristics, which allows us to mimic the basic features of neuromorphic circuits integrating functions of image sensing, memory and processing into the same space in the device. When an optical pulse is applied to the phototransistor, the trapped photogenerated electrons inside the perovskite CsPbBr₃-QD layer take a long time to decay. In order to obtain a rapid decay of the photocurrent, applying an electric field reset pulse can induce many positive

carriers in the CNT film and recombine with the trapped electrons that escape to the CNT layer, that is, the electrical reset pulse will accelerate the detrapping process.

Corresponding modifications have been made in Supplementary Fig. 19 in the revised Supplementary Information.

18. [IMPORTANT] I am not sure that the detectivity values provided by the authors are actually valid. There is a big miss-interpretation in the 2D community where the formula for shot-noise limited detectors is always used despite the fact that this might not be the case. See: <https://www.nature.com/articles/s41566-018-0288-z>, <https://www.nature.com/articles/s41467-018-07643-7>, I urge the authors to address this point.

Response:

Thanks for your valuable comments. Indeed, the characteristics of noise power density are very important for comprehensive and accurate device performance evaluation. As the reviewer pointed out, for the devices made of recent emerging semiconductors (such as perovskites, 2D materials or organics), sometimes the noise density was underestimated and therefore the detectivity of device was overestimated. In this case, it is assumed that the noise power density is independent of frequency and equal to shot-noise, but it is actually NOT correct.

However, in our original submission, we have performed noise measurements and extracted dark noise currents for a correct evaluation of device performance. Figure R26 shows the measured noise power density ($\sqrt{S_n}=1.14 \times 10^{-11} \text{ A/Hz}^{1/2}$), which

indicating that the $1/f$ (flicker) noise is dominant, and the specific detectivity of our phototransistor was evaluated to be 2×10^{16} Jones. On the other hand, if only the shot-noise is used for evaluation, ($\sqrt{2eI_{\text{dark}}}=1.79 \times 10^{-14}$ A/Hz $^{1/2}$), the obtained detectivity (1.3×10^{19} Jones) will be overestimated by three orders of magnitude higher than the actual situation. Therefore, we state that, in our research, the evaluation of device performance parameter is based on scientific judgment.

Figure R26 Noise power density (S_n) of the phototransistor showing a strong $1/f$ component measured at a frequency of 1 Hz. $S_n = 1.3 \times 10^{-22}$ A 2 /Hz. $V_{\text{DS}} = 1$ V, $V_{\text{GS}} = 5$ V.

19. Figure 3, more explanation is required about when pulse numbers are increased the image become clearer.

Response:

According to the reviewer’s suggestion, we added more explanation about the relationship between pulse number and image quality. It was found that the on-current of a single phototransistor increased steadily with an increase in the pulse number of light (Fig. 2f), and the pulse number was reduced for a higher lighting power density to

achieve the target synaptic weight, thereby representing the persistent strengthening of synaptic characteristics. The on-currents of 1024 pixels in the sensor array have excellent uniformity, which enables high-quality image sensing, and the larger on-current represents a deeper impression in the evolution of learning and training of images, which allows us to demonstrate the function of neuromorphic pattern reinforcement. This behavior is similar to human vision, where the features of familiar faces are clearer than the features of a strange faces occasionally seen. After training 0, 10, 20, 50, 100 and 200 pulses with an ultra-weak light ($1 \mu\text{W}/\text{cm}^2$), the weight map of the sensor array obtained shows different resemblance degrees as well as image sharpness to the input number "8" pattern (Fig. 3e). Therefore, by training a highly integrated sensor array with weak light pulses, the function of neuromorphic reinforcement learning has been demonstrated experimentally.

Corresponding modifications have been made on Page 10 (Lines 20-25) and Page 11 (Lines 1-11) in blue in the revised manuscript.

REVIEWER COMMENTS

Reviewer #1 (Remarks to the Author):

The authors have addressed all the issues and this paper can be accepted by nature communications now.

Reviewer #2 (Remarks to the Author):

The authors have fully addressed the questions raised from the reviewers. The large-scale integration of CNT devices is very impressive. I would recommend the acceptance by Nature Communications.

Reviewer #3 (Remarks to the Author):

The authors have responded most of the questions well. I recommend this manuscript to be published in Nature Communications after minor revision as given below:

1. In Figure 3 (Revised manuscript), it is recommended to provide the impact of different light power on image quality.
2. In Supplementary Figure 20, why decay time is almost independent of power density but rise time is strongly dependent on power density? Also explain why rise time is higher than decay time?
3. The explanation regarding the mechanism of effective splitting of exciton when charges are transferred between CNTs and CsPbBr₃-QDs is still unsatisfactory. The authors seem to have provided a reference which is not compatible. Please address this issue appropriately.

Responses to the comments of reviewers

Reviewer #1:

The authors have addressed all the issues and this paper can be accepted by nature communications now.

Response:

Thank you very much for your review.

Reviewer #2:

The authors have fully addressed the questions raised from the reviewers. The large-scale integration of CNT devices is very impressive. I would recommend the acceptance by Nature Communications.

Response:

Thank you very much for your positive comments.

Reviewer #3:

The authors have responded most of the questions well. I recommend this manuscript to be published in Nature Communications after minor revision as given below:

Response:

Thank you very much for your positive comments.

1. In Figure 3 (Revised manuscript), it is recommended to provide the impact of different light power on image quality.

Response:

According to your valuable suggestion, we have measured the weight map of the sensor array after training with 10 pulses under a 405 nm light with various lighting power densities (Fig. R1). The weight map shows different resemblance degrees and image sharpness compared with the input number “8” pattern, indicating that higher lighting power density can speed up the pattern learning process.

Figure R1 Measured training weight results of the sensor array after training with 10 pulses under a 405 nm light with various lighting power densities of 4.0 $\mu\text{W}/\text{cm}^2$, 0.3 mW/cm^2 , 1.0 mW/cm^2 , 2.5 mW/cm^2 and 4.0 mW/cm^2 (pulse width, 250 ms; pulse interval, 250 ms).

Corresponding modifications have been made on Page 11 (Lines 9-12) and Page 13 (Lines 2-5) in blue, and in Fig. 3f in the revised manuscript.

2. In Supplementary Figure 20, why decay time is almost independent of power density but rise time is strongly dependent on power density? Also explain why rise time is higher than decay time?

Response:

Thanks for your valuable comments. We would like to emphasize the mechanisms

that affect the rise and decay times of device are different. The current of the phototransistor will increase if a light with an enhanced lighting power density is applied to the device, thereby generating more photo-generated electrons per unit time to be trapped in CsPbBr₃-QDs, therefore, the rise time is strongly dependent on the lighting power density. On the other hand, a reset pulse is applied to the transistor when the light is turned off, leading to a recombination of the holes in the CNT channel and the electrons trapped in the CsPbBr₃-QDs (Supplementary Figure 19), which significantly accelerates the detrapping process. Therefore, the decay time is mainly determined by the reset pulse and is independent of the lighting power density.

3. The explanation regarding the mechanism of effective splitting of exciton when charges are transferred between CNTs and CsPbBr₃-QDs is still unsatisfactory. The authors seem to have provided a reference which is not compatible. Please address this issue appropriately.

Response:

Thanks for your valuable comments. As shown in Figs. R2a and R2b, when CsPbBr₃-QDs are spin-coated on the CNT channel, the carriers (electrons) flow from CsPbBr₃-QDs to CNTs due to the difference of the Fermi levels, thereby leading to a built-in electric field in the space charge region, where the CsPbBr₃-QD side has a higher voltage potential. This built-in electric field is the key to effectively splitting photo-generated carriers. According to the reviewer's comments, we have corrected the references to prove this point (G. Konstantatos, et al., Nat. Nanotechnol., 7, 363-368,

2012; Z. Sun, et al., Adv. Mater. 24, 1202220, 2012; D. Kufer, et al., ACS Photonics, 3, 2197-2210, 2016). Due to the opposite electric force applied on them, the photo-generated electrons and holes formed in the space charge region will be separated by the built-in electric field. The holes flow to the CNT side with a lower voltage potential while the electrons flow to the CsPbBr₃-QD side with a higher voltage potential and are trapped (Fig. R2b). This process can also be verified by the negative shift of the I_{DS} - V_{GS} characteristics (Fig. R2c) and the transient PL spectra of CsPbBr₃-QD and CNT/CsPbBr₃-QD films (Fig. R2d). When CsPbBr₃-QDs are spin-coated, the negative shift of the I_{DS} - V_{GS} characteristics indicates the p-type CNT channel is more difficult to be turned on because the electrons flow from the CsPbBr₃-QD to the p-type CNT and are partially depleted.

Figure R2 The mechanism of photo-generated carrier separation. (a) Band structure of

CNT and CsPbBr₃-QD. (b) Energy band diagram under dark (top panel) and illuminated (bottom panel) conditions. (c) I_{DS} - V_{GS} characteristics of the transistors with CNT channel and CNT/CsPbBr₃-QD channel under dark condition. (d) Transient PL spectra of CsPbBr₃-QD and CNT/CsPbBr₃-QD films.

Corresponding modifications have been made in Supplementary Fig. 11 in the revised Supplementary Information.

REVIEWERS' COMMENTS

Reviewer #3 (Remarks to the Author):

The authors have addressed all the issues. I recommend this paper to be published by nature communications.